# Fracture healing on non-union fracture model promoted by non-thermal atmospheric-pressure plasma

Kosuke Saito[1], Hiromitsu Toyoda[1,2]*, Mitsuhiro Okada[1,2], Jun-Seok Oh[3], Katsumasa Nakazawa[1], Yoshitaka Ban[1], Kumi Orita[2], Akiyoshi Shimatani[4], Hana Yao[2], Tatsuru Shirafuji[3], Hiroaki Nakamura[1,2]

1 Department of Orthopedic Surgery, Graduate School of Medicine, Osaka City University, Osaka, Japan,
2 Department of Orthopedic Surgery, Graduate School of Medicine, Osaka Metropolitan University, Osaka, Japan, 3 Department of Physics and Electronics, Graduate School of Engineering, Osaka Metropolitan University, Osaka, Japan, 4 Department of Orthopedic Surgery, Saiseikai Nakatsu Hospital, Osaka, Japan

* h-toyoda@omu.ac.jp

**Data Availability Statement:** All relevant data are within the paper and its Supporting Information files.

## Abstract

Non-thermal atmospheric-pressure plasma (NTAPP) is attracting widespread interest for use in medical applications. The tissue repair capacity of NTAPP has been reported in various fields; however, little is known about its effect on fracture healing. Non-union or delayed union after a fracture is a clinical challenge. In this study, we aimed to investigate how NTAPP irradiation promotes fracture healing in a non-union fracture model and its underlying mechanism, in vitro and in vivo. For the in vivo study, we created normal and non-union fracture models in LEW/SsNSlc rats to investigate the effects of NTAPP. To create a fracture, a transverse osteotomy was performed in the middle of the femoral shaft. To induce the non-union fracture model, the periosteum surrounding the fracture site was cauterized after a normal fracture model was created. The normal fracture model showed no significant difference in bone healing between the control and NTAPP-treated groups. The non-union fracture model demonstrated that the NTAPP-treated group showed consistent improvement in fracture healing. Histological and biomechanical assessments confirmed the fracture healing. The in vitro study using pre-osteoblastic MC3T3-E1 cells demonstrated that NTAPP irradiation under specific conditions did not reduce cell proliferation but did enhance osteoblastic differentiation. Overall, these results suggest that NTAPP is a novel approach to the treatment of bone fractures.

## Introduction

Although most bone fractures heal, approximately 5–10% of fractures experience incomplete healing, leading to delayed union or non-union [1, 2]. Delayed unions and non-unions can result in significant clinical complications, persistent pain, and psychological burdens, thereby affecting the quality of life in the long term [3]. Treating these conditions is challenging and poses a considerable economic burden owing to indirect costs such as decreased productivity [4].

**Funding:** This work was supported by JSPS KAKENHI number JP19K03811 and the Glocal hub of wisdom and wellness filled with smiles. The funders had no role in study design, data collection and analysis, decision to publish, or preparation of the manuscript. Initial of the authors who received JSPS KAKENHI: JO URL: https://www.jsps.go.jp/j-grantsinaid/ Initial of the authors who received the Glocal hub of wisdom and wellness filled with smiles: HT.

**Competing interests:** The authors have declared that no competing interests exist.

Several bone healing strategies have emerged in surgery, such as dynamization, masquelet technique, and bone transport [5–7]. Additionally, low-intensity pulsed ultrasound (LIPUS) and extracorporeal shock wave therapy (ESWT) have been explored as potential adjunctive therapies for accelerating fresh fracture healing [8, 9]. The transplantation of autogenous bone is regarded as the gold standard, but its limitations include low volume and donor site morbidity. Therefore, the use of growth factors such as bone morphogenetic proteins (BMPs), epidermal growth factor (EGF), platelet-rich plasma (PRP), stromal vascular fraction, and peptides show promise in bone healing [10–15]. PRP recruits mesenchymal stem cells, promotes angiogenesis, and has the potential to aid in fracture response [16]. Stromal vascular fraction, rich in stem cells and growth factors, has shown positive results in combination with PRP for bone healing [14]. Although PRP research holds promise, it currently lacks standardization. Furthermore, it remains unclear which specific PRP types can yield superior outcomes in terms of vital new bone formation [17, 18]. BMPs stimulate osteoblasts, and bone tissue engineering techniques using suitable scaffolds combined with BMP become new options in reconstructive bone surgery. However, their clinical use faces limitations such as soft-tissue swelling, ectopic bone formation, and high costs [13, 19, 20]. While EGF has been shown to promote cell proliferation, migration, angiogenesis, and osteogenesis [15], conflicting reports on its role in bone formation may be attributed to inconsistencies in concentration and administration [21, 22]. Parathyroid hormone peptide (PTHP) is an anabolic bone therapeutic medicine used to treat osteoporosis. Studies have shown that PTH could improve fracture healing at different skeletal sites [23–28]. The paradigm shift from transplantation of autogenous bone to bone tissue engineering seems promising, and the search for better and more consistent treatment options continues. Recently, non-thermal atmospheric-pressure plasma (NTAPP) has attracted tremendous interest in the biomedical field. NTAPP generates various reactive oxygen and nitrogen species (RONS), charged particles, and UV photons [29, 30]. The effects of NTAPP on biological tissues have been demonstrated at various levels, including sterilization [31–34], wound healing [35, 36], increased cell migration [37], and cancer therapy [38]. Among the biomedical applications mentioned above, we focused on regenerative medicine using NTAPP, such as wound healing, skin regeneration, and bone regeneration. Our previous study by Shimatani et al. reported how NTAPP affected bone regeneration using a rabbit model with a critical-sized ulnar bone defect was used [39].

In this study, we focused on quality of life to improve the completeness of bone healing. Here, we use rats with fracture models such as normal fracture models or non-union fracture models. Also, we investigate the potential of NTAPP in fracture healing using in vivo and in vitro studies to simulate clinically relevant conditions.

## Materials and methods

### Helium microplasma jet treatment

The plasma jet apparatus used in this study was manufactured in-house, as described in the previous studies [39]. The plasma jet consists of a 150 mm long glass tube tapering to 0.65 mm at the nozzle tip. A sinusoidal high voltage with an amplitude of approximately 10 kV and a frequency of 33 kHz was applied to a single external ring copper electrode positioned 50 mm away from the nozzle outlet. In this configuration, known as a capillary dielectric barrier discharge plasma jet, both positive and negative discharges are alternately generated. Helium (He) gas was utilized as the primary discharge gas because it produces a stable glow discharge in the ambient air. The flow rate of He through the nozzle was set to 1.5 standard liters in the in vivo study and 1.0 standard liter per minute in the in vitro study.

### In vivo study

**Animal models.** 8-week-old male LEW/SsNSlc in-bred rats (Japan SLC Inc., Hamamatsu, Japan) were used for this animal experiment. All rodents were maintained in a 24-hour light and dark cycle, with ad libitum access to food and water. All experimental procedures were approved by the Osaka City University, currently Osaka Metropolitan University, Graduate School of Medicine Committee on Animal Research (approval number: 21092). We created a normal fracture model for a preliminary experiment, followed by a non-union fracture model. The rodents were monitored daily for indications of dehydration, malnutrition, pain, infection, and abnormal behavior. The rats were eventually sacrificed by $CO_2$ asphyxiation. Twenty-four rats were assigned to the normal fracture model, 20 rats to radiological and histological evaluation of the non-union fracture model, and 14 rats to biomechanical evaluation. No rats were excluded due to adverse events or unexpected deaths during the study period.

**The normal fracture model.** The rats were anesthetized by subcutaneous injection of ketamine (50 mg/mL) (Daiichi Sankyo Healthcare Co. Ltd, Tokyo, Japan) and xylazine (0.2 mg/mL) (Bayer HealthCare, Leverkusen, Germany) at a 10:3 ratio at a dose of 1 mL/kg total body weight. This model was developed by improving upon previously reported methods [40–43]. To induce a fracture, a transverse osteotomy was performed in the middle of the right femoral shaft using a bone saw, following a 30 mm incision made on the hind limb in the rats, as shown in Fig 1A. A 1.2-mm Kirschner wire was retrogradely inserted from the fossa intercondylaris, penetrating through to the greater trochanter. The distal end of the wire was embedded into the articular surface of the knee joint. In line with previous studies using a similar model [41, 42], weight-bearing was allowed immediately after surgery without immobilization. In this preliminary study, the rats were divided into three groups based on the duration of NTAPP treatment: 0 min (control group), 5 min, and 15 min. All the NTAPP treatments were conducted at around 10 mm away from the nozzle exit. NTAPP treatment was administered only once during surgery, as shown in Fig 1B. Each group was comprised of eight rats.

**The non-union fracture model.** A non-union fracture model was developed by modifying the technique proposed by Kokubu et al. [43]. The periosteum surrounding the fracture site was cauterized with a 2 mm margin on either side after creating a normal fracture model, as shown in Fig 1C. In this study, two groups were defined based on the period of treatment received. The control group was exposed only to the helium gas around the fracture site for 5 min, whereas the NTAPP-treated group was exposed to NTAPP for the same 5 min duration (Fig 1D). Each group consisted of ten rats. The distance from the nozzle and the treatment duration were similar to that used for the normal fracture model. We positioned this as our main experiment and evaluated the treatment effect.

**X-ray analysis.** Radiography was performed postoperatively and evaluated using a previously reported scoring system based on cortical re-bridging and healing acceleration (Table 1) [44]. A radiographic scoring scale was used to evaluate the fracture union. Radiographic scoring was conducted by two authors (K. N. and K. O.) in a blinded manner. Radiography was performed at 2 and 4 weeks after the surgery in the normal fracture model and at 2, 4, and 8 weeks after the surgery in the non-union fracture model, respectively.

**Histological analysis.** The femoral bone fractures in each group were harvested for histological evaluation after sacrifice. The soft tissues surrounding the fracture site were dissected without excising the callus, and the Kirschner wire was carefully removed without any destruction of the callus. Each specimen was decalcified using 10% Ethylenediaminetetraacetic acid (Muto Pure Chemicals Co. Ltd., Tokyo, Japan), dehydrated using an alcohol series, and embedded in paraffin. Sections were cut at a thickness of 4 mm and stained using Masson's trichrome stain and Safranin-O/fast green (Safranin-O). Staining was performed based on the

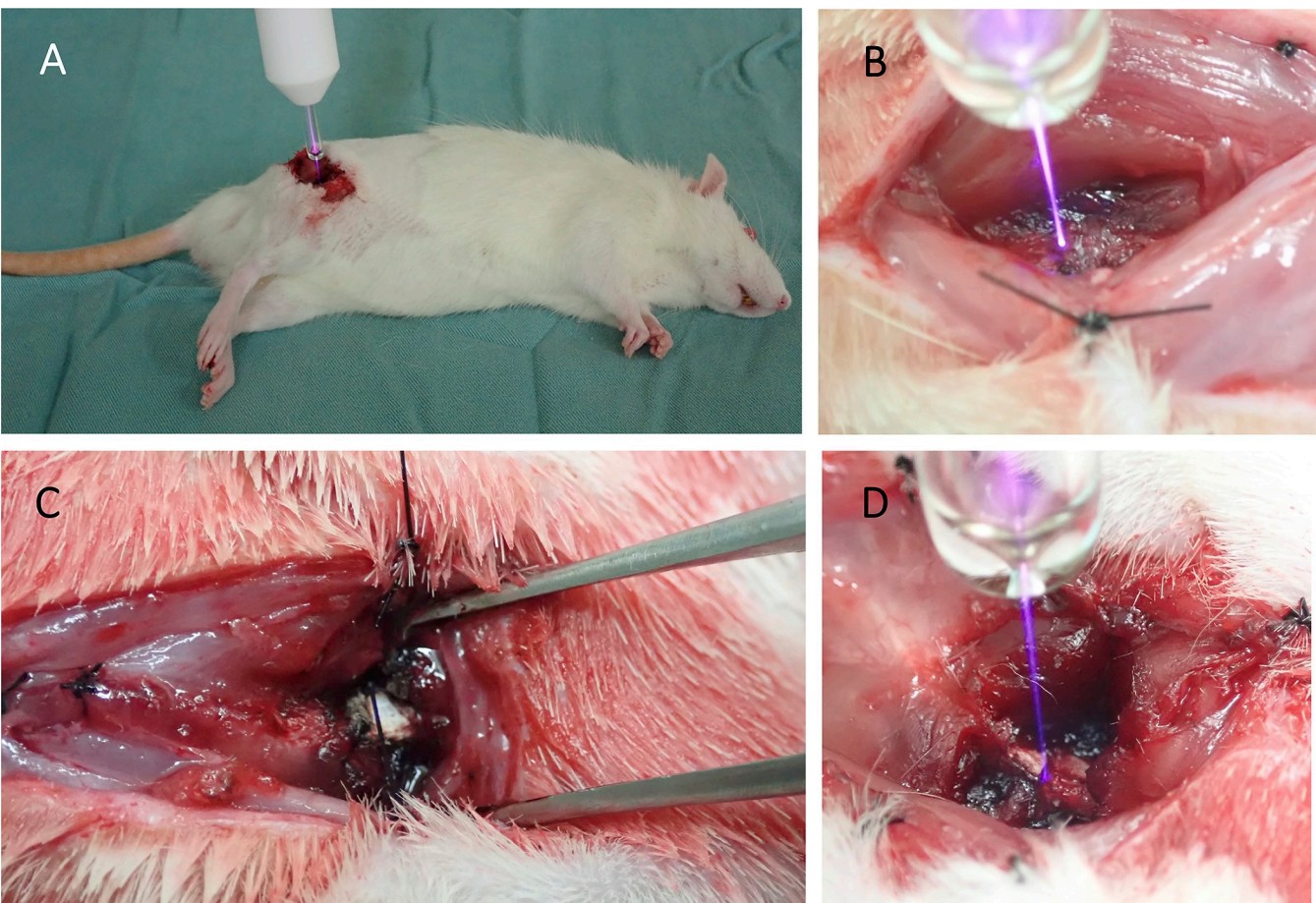

**Fig 1. A** shows a plasma treatment on the rat model with either the normal or non-union fracture model during the surgery. **B** shows the fracture site of a normal fracture model during the plasma treatment. **C** shows a non-union fracture model which was thermally damaged on the periosteum. **D** shows a plasma treatment on the non-union fracture model during the surgery.

standard protocol. The sections were observed under a microscope (BX53F; Olympus, Tokyo, Japan), and all images were captured by a digital camera (DP74, Olympus, Tokyo, Japan). Histological evaluation was performed at 2 and 4 weeks after the surgery for the normal fracture model in the preliminary study and at 2, 4, and 8 weeks after the surgery for the non-union fracture model in the main study, respectively.

**Table 1. Radiographic scoring scale.**

|  | Points |
|---|---|
| No bridging, no callus formation | 0 |
| No bridging, initiation of a small amount callus | 1 |
| No bridging, obvious initial callus formation near fracture | 2 |
| No bridging, marked callus formation near and around fracture site | 3 |
| Rebridging of at least one of the cortices, marked callus formation near and around fracture site | 4 |
| Rebridging of at least one of the cortices, marked and complete callus formation around fracture site | 5 |
| Rebridging of both cortices, and/or some resolution of the callus | 6 |
| Clear rebridging of both cortices and resolution of the callus | 7 |

**Micro-computed tomography analysis.** Rats in each group of the non-union fracture model were sacrificed 8 weeks after the surgery, and the Kirschner wire was carefully removed without disrupting the callus. The extracted rat femoral bones were subjected to micro-computed tomography (μ-CT) (SMX-90CT Plus inspeXio; Shimadzu Co., Kyoto, Japan) after the neutral buffered solution fixation process at room temperature. As described previously, the reconstructed scanning data were used to quantify the new bone volume using a 3D image processing software (ExFact VR; Nihon Visual Science Inc., Tokyo, Japan) [39, 45].

**Biomechanical assessment.** Seven rats in each group of the non-union fracture model were sacrificed 8 weeks after the surgery. A standardized 3-point bending test was performed on each group using a bending tester (EZ Graph, Shimadzu, Kyoto, Japan) [46, 47]. Two parameters, the maximum load (N) and failure energy (mJ), were used to assess the strength at the fracture site.

**XPS.** To identify surface chemistry changes in the rat femoral bone cortex, X-ray photoelectron spectroscopy (XPS) was performed. XPS analysis was conducted in a laboratory using small femoral bone tips (approximately 3 mm × 3 mm). XPS measurement for the same sample was performed twice. First, the femoral bone sample without NTAPP irradiation was analyzed using XPS (ESCA-3400, Shimadzu, Kyoto, Japan). Subsequently, the same bone sample was re-analyzed by XPS after irradiation with NTAPP for 5 min on site.

**Surface wettability.** The water contact angle (WCA) was measured by a contact angle analyzer (DMe-211, Kyowa Interface Sci. Co., Ltd., Tokyo, Japan) to investigate the surface energy change after the exposure of NTAPP. Deionized water (1 mL) was dropped onto a mid-shaft area of the femoral bone. The bone surface was either left untreated or treated with NTAPP for 5 min.

## In vitro study

**Cell culture.** The MC3T3-E1 murine pre-osteoblast cell line was purchased from the RIKEN Cell Bank. MC3T3-E1 cells were grown at 37˚C in a moist atmosphere of 5% $CO_2$ in α-minimal essential medium (α-MEM, Wako Pure Chemical Co., Osaka, Japan) supplemented with 10%(v/v) fetal bovine serum (FBS, Sigma-Aldrich, St. Louis, USA) and 1%(v/v) penicillin-streptomycin (Wako Pure Chemical, Osaka, Japan).

**NTAPP irradiation.** The plasma jet apparatus was the same as in the animal study above. The distance between the nozzle and the surface of the cell culture medium was fixed at 15 mm. NTAPP was directly irradiated into each solution in each well of a 24-well plate for each experiment. Each experiment was conducted in the presence of 500 μL of medium to avoid cellular desiccation in 24-well plates.

**Cell proliferation.** To assess the effect of direct irradiation with NTAPP on the proliferation of MC3T3-E1 cells, an MTT assay was conducted for each experimental condition using an MTT Cell Proliferation/Viability Assay kit (R&D Systems, Minneapolis, USA) following the manufacturer's specifications. Briefly, $1.8 \times 10^5$ cells from cell culture passage five were seeded onto a 24-well plate with-MEM containing 10% FBS. After cultivation for 24 h to allow for sufficient cell attachment, the medium was replaced, and NTAPP was irradiated to the fresh medium surface. NTAPP irradiation was conducted once, and irradiation duration was relatively shorter at 5, 10, and 15 s, compared to the animal study, which was 5 min and 15 min. Two control groups were defined: untreated and irradiated with helium gas for 5 s. After incubating for 24 h and 48 h, the MTT assay was performed. The absorbance was measured at a fixed wavelength of 570 nm using a microplate reader (U-3000 spectrophotometer; Hitachi, Tokyo, Japan). Each group was assessed in triplicates.

**Alkaline phosphatase (ALP) activity assay.** An ALP activity assay was conducted to assess the effect of direct irradiation with NTAPP on the osteoblastic differentiation of MC3T3-E1 cells. Briefly, $1.0 \times 10^5$ cells from cell culture passage five were seeded onto a 24-well plate with MEM containing 10% FBS. After cultivating for 24 h to acquire sufficient cell attachment, the medium was changed to α-MEM containing 10% FBS and osteoblast-inducer reagents (OIR; Takara Bio Inc., Otsu, Japan), including L-ascorbic acid, hydrocortisone, and β-glycerophosphate. The medium was refreshed every two days, and an ALP activity assay was performed on the 6th day after the initial NTAPP irradiation. A negative control group was defined as the group cultured in MEM, and a positive control group was defined as the group cultured in OIR. None of the control groups received NTAPP irradiation. The Pierce BCA Protein Assay Kit (Thermo Scientific, Rockford, USA) was used to measure the protein concentration in each well. ALP activity was calculated based on the instructions of the Lab-Assay ALP (Wako Pure Chemical Inc., Osaka, Japan), and the absorbance was measured at the wavelength of 405 nm using a microplate reader. Each group was assessed in triplicates.

## Statistical analysis

All data represent the mean ± standard deviation (SD). In the in vivo study, the Kruskal-Wallis test was used for multiple comparisons of radiographic scores, and the Mann–Whitney U test was used to compare the biomechanical assessment and new bone volume in the control and NTAPP-irradiated groups. Radiographic scores were assessed using intraclass correlation coefficients (ICC) interpreted as very good if 0.81–1.00, good if 0.61–0.80, moderate if 0.41–0.60, fair if 0.21–0.40, and poor if < 0.20 [48]. One-way analysis of variance with a post hoc t-test was used to compare differences in cell viability and ALP activity assays. Statistical significance was set at $P < 0.05$. All analyses were performed using the EZR software (version 2.7–2; Saitama Medical Center, Jichi Medical University, Shimotsuke, Japan).

# Results

## Radiological analysis of the normal fracture model

All rats having a normal fracture model demonstrated progressive recovery of the fracture site over time, as evidenced by X-ray imaging (Fig 2A and 2B). The X-ray scores obtained at 2 and 4 weeks after surgery did not show any significant difference between the 5-minute irradiation group (3.3 ± 2.0, 4.9 ± 1.0), the 15-minute irradiation group (4.5 ± 1.9, 4.6 ± 0.9), and the control group (3.0 ± 1.8, 4.7 ± 0.9) (Fig 2C). The ICC of the radiographic scores was 0.833, which is considered to be reliable.

## Histological analysis of the normal fracture model

Callus formation was observed around the fracture site at 2 weeks after the surgery in all three groups, with irradiation times ranging from 0–15 mins. Safranin-O staining illustrated the process of endochondral ossification (Fig 2D–2G). Endochondral ossification is a bone formation process in which the cartilage scaffold is gradually replaced by bone. Delayed fracture healing due to NTAPP irradiation was not observed in any sample.

## Surface chemistry and energy of rat femur

Fig 3 shows the typical XPS spectra of plasma-treated and untreated rat femurs. The XPS results showed a significant: relatively higher intensity of the O1s peak at 533 eV and a smaller intensity of C1s peak at 286 eV after 5 min of NTAPP irradiation (Fig 3A) compared to the untreated surface. It is clearly shown that the O1s/C1s ratio increased after NTAPP treatment

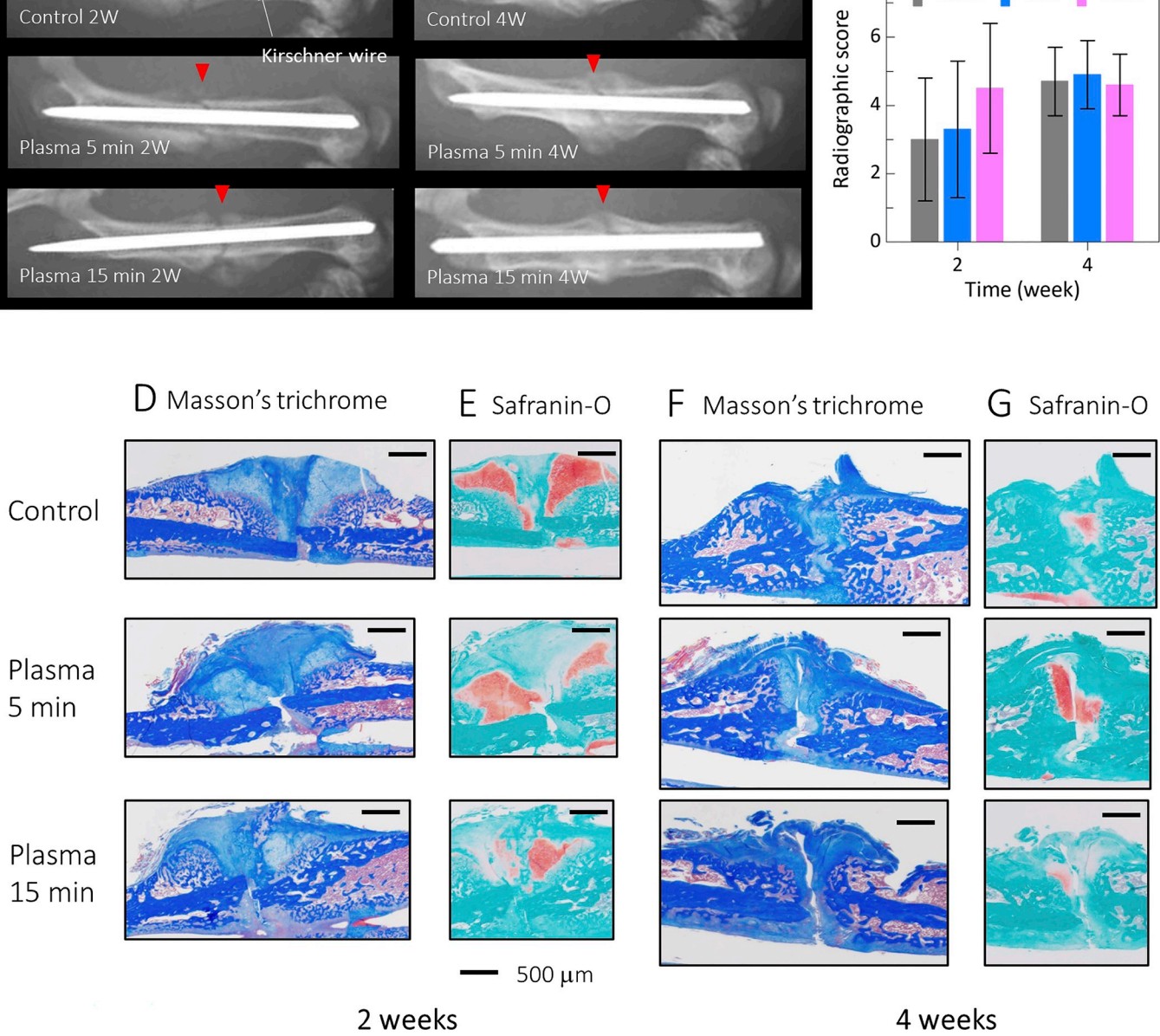

**Fig 2. A** shows X-ray images of normal fracture models with and without plasma treatment after 2 weeks of the surgery. **B** shows X-ray images of normal fracture models with and without plasma treatment after 4 weeks of the surgery. **C** shows the summary of the radiographic scores of the X-ray images; the 5-minute irradiation group (3.3 ± 2.0, 4.9 ± 1.0), the 15-minute irradiation group (4.5 ± 1.9, 4.6 ± 0.9), and the control group (3.0 ± 1.8, 4.7 ± 0.9). Tissue sections of normal fracture models **D** shows Masson's trichrome staining, and **E** shows Safranin-O staining at 2 weeks after the surgery. Tissue sections of normal fracture models **F** show Masson's trichrome staining and **G** shows Safranin-O staining at 4 weeks after the surgery. The scale bar indicates 500 µm.

(pristine; 44.6 ± 10, NTAPP treated; 120.9 ± 110.2) (Fig 3B). It seems that the O1s/C1s increase may be due to the oxidization by the reactive oxygen species which was generated by an interaction between the plasma species (ionized gas) and ambient air. The bone surface was oxidized after plasma treatment for 5 min.

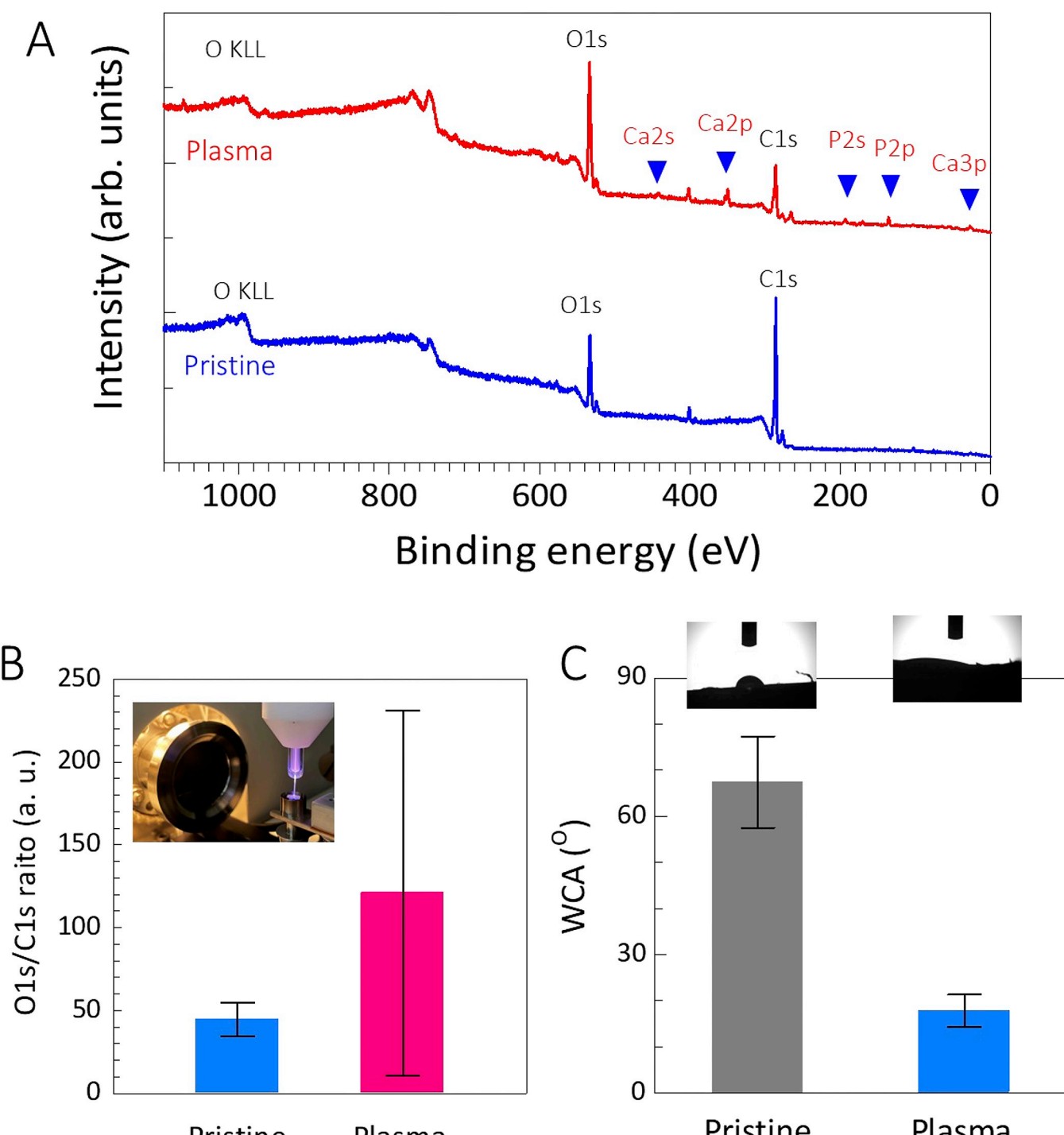

**Fig 3.** **A** shows a comprehensive XPS spectrum scan conducted on-site, comparing plasma-treated samples with untreated controls (pristine). Following plasma treatment, O1s levels significantly increased, accompanied by a reduction in C1s levels. Furthermore, basic elements of phosphorus and calcium appeared following the plasma treatment. **B** shows the O1s/C1s ratio of pristine and plasma treatment. **C** shows water contact angles of pristine and plasma treatment, indicating the bone surface to be hydrophilic after the plasma treatment.

The WCA of 17.8° ± 3.5 for the NTAPP-treated femur bone surface was measured. It is much lower than that of the untreated femoral bone surface, which was 67.4° ± 9.9 (Fig 3C). Together, NTAPP irradiation for 5 min affected the surface chemistry of the bone surface and surface energy.

## X-ray imaging analysis of the non-union fracture model

The X-ray examination results for the non-union fracture model are shown in Fig 4. The NTAPP-treated group showed considerable callus formation around the fracture site at 4 weeks after the surgery. The calluses connect both ends of the fracture gap and fill the cortical gaps on both sides. The callus was consolidated 8 weeks postoperatively, and the remodeling process commenced. In contrast, the control group showed no obvious callus around the fracture sites during the observation period of up to 8 weeks. At 8 weeks, the edges of the fracture site appeared atrophic, resulting in a non-union appearance.

Fig 4C shows the radiographic fracture scores for each period. 2 weeks after the surgery, there were no significant differences between the control (0.5 ± 0.67) and NTAPP-treated groups (1.0 ± 0.77), respectively. Of course, the scores are much smaller than those of the normal fracture models, as shown in Fig 2C. These scores do not increase much even after 8 weeks in the case of control. This reflects well the severe conditions of the non-union fracture model. At 4–8 weeks after the surgery, consistent with the imaging results, the NTAPP group showed significantly improved scores of 3.6±0.66, $P < 0.01$ at 4 weeks and 5.5±1.6, $P < 0.01$ at 8 weeks, compared to the control group 0.8 ± 0.6 at 4 weeks and 1.2 ± 1.0 at 8 weeks, respectively. The ICC of the radiographic scores was 0.823, which was considered reliable.

## Histological analysis of the non-union fracture model

In the control group, Masson's trichrome staining revealed that the fracture gap was filled with fibrous tissue. The border between the fractured bone and fibrous tissue was clearly observed in any cases 2, 4, and 8 weeks after the surgery (Fig 5A). Conversely, in the NTAPP-treated group (Fig 5C), bridging callus around the fracture site was observed 4 weeks after the surgery. Furthermore, the fracture gap was nearly bridged by abundant calluses at 8 weeks after the surgery.

Histological examination with Safranin-O staining was also demonstrated. In the control group, very few chondrogenic lesions were observed around the fracture site within 8 weeks. The presence of only a small number of chondrogenic lesions in the non-union fracture model clearly indicates its distinctiveness from the normal fracture model in Fig 5B. This indicates that the process of healing the fracture has not made significant progress. The NTAPP-treated group showed improved endochondral ossification 4 weeks after the operation (Fig 5D). At 8 weeks, it is significant that the cartilage-forming area began to disappear slightly, and the cartilage-forming area was replaced by newly formed bone. This is clear evidence that the bone union has progressed in the non-union fracture model after the single plasma treatment with NTAPP irradiation for 5 min at the surgery. Taken together, endochondral ossification clearly started at 4 weeks, and bone healing progressed through 8 weeks in the NTAPP group, whereas bone formation was not observed in the control group.

## Micro-CT and biomechanical analysis

Fig 6 shows the representative 3D-reconstructed CT images of the femoral bone and the volume of new bone at the fracture site. The NTAPP-treated group contained significantly more volume of new bone (98.2 ± 30.3 mm³, $P < 0.01$) than the control group (38.7 ± 11.4 mm³, $P < 0.01$), respectively, in Fig 6D.

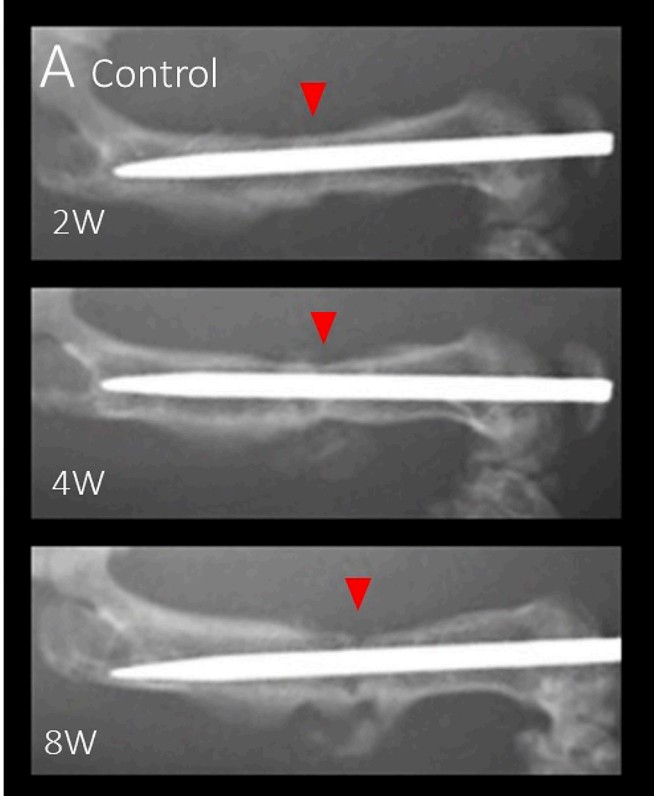
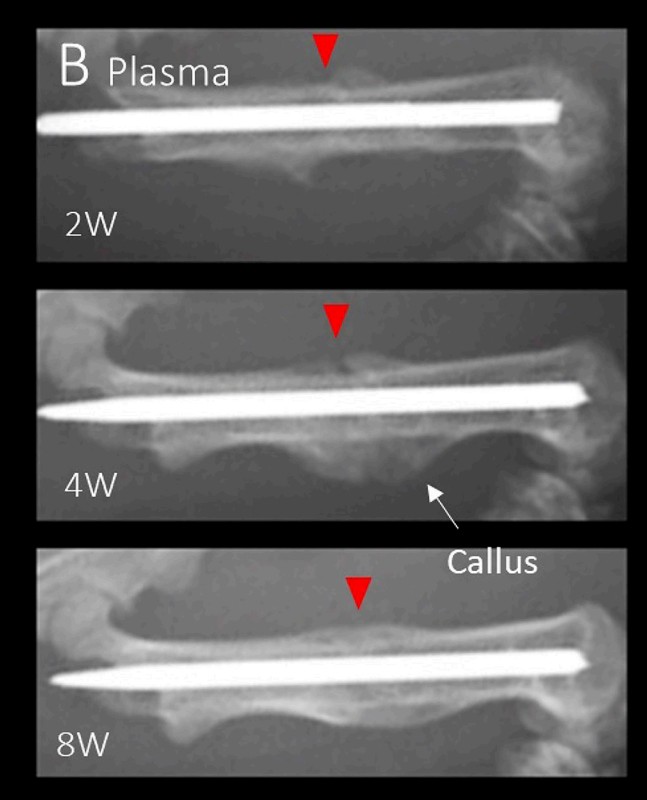

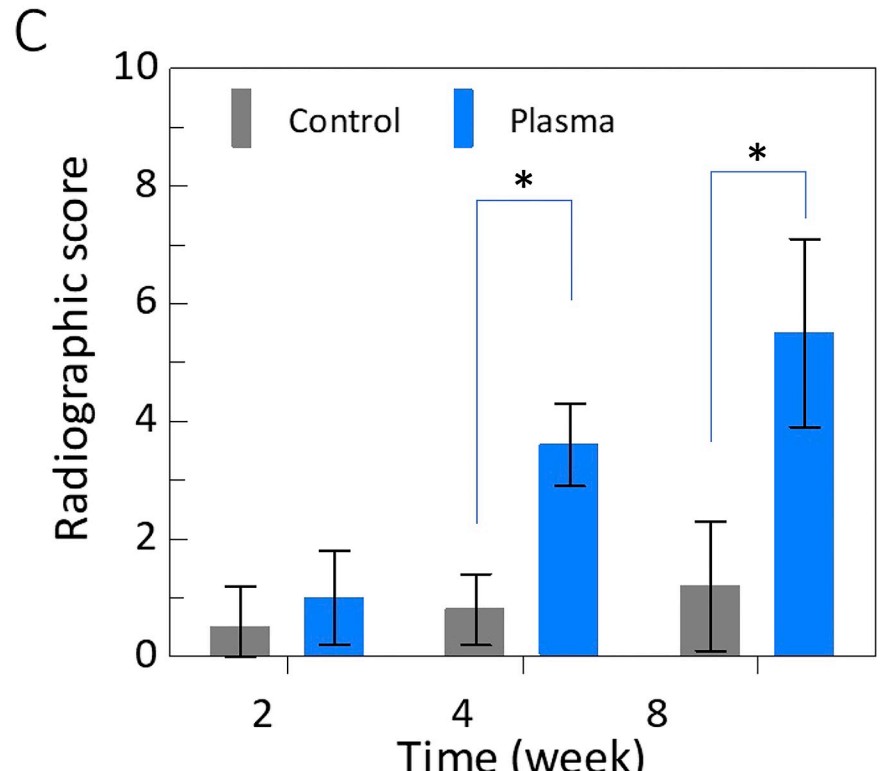

**Fig 4.** **A** shows X-ray images of the non-union fracture model, which are untreated controls 2, 4, and 8 weeks after the surgery. **B** shows X-ray images of the non-union fracture model with plasma treatment after 2, 4, and 8 weeks of the surgery. **C** shows the summary of the radiographic scores of the X-ray images.

Moreover, the maximum load in Fig 6E and failure energy in Fig 6F were significantly higher in the NTAPP-treated group compared to the control group. These results reflect well the strong correlation between the mechanical property and the fracture healing progress.

## Effect of NTAPP on MC3T3-E1 cell proliferation

MC3T3-E1 cell viability was determined after incubation for 24 and 48 h (n = 6 for each group). The viability of cells treated with NTAPP irradiation for 5, 10, and 15 s and with 5 s of irradiation with He gas or no treatment was not significantly different (Fig 7A). We confirmed that NTAPP irradiation for 5–15 s did not have any harmful effects on the cells compared to the controls. Based on these findings, we investigated the effect of this dose on osteoblast differentiation.

## Osteoblastic differentiation after irradiation of NTAPP

The ALP activity in NTAPP-treated MC3T3-E1 cells was measured after incubating for 6 days, as shown in Fig 7B. ALP activity in MC3T3-E1 cells is increased in all cases of NTAPP irradiation. The activity was significantly higher after irradiation compared to the positive control (OIR). Regarding the effect of NTAPP irradiation, ALP activity did not differ significantly in response to the differing duration of NTAPP irradiation.

## Discussion

Non-thermal atmospheric-pressure plasma (NTAPP) has been used in medical treatments, and its effect on wound healing [35, 36], hemostasis [49], sterilization [31–34, 50], and cancer treatment [51–53] have been previously reported. Shimatani et al. reported the effects of NTAPP on bone regeneration in a rabbit model of an ulnar bone critical-size defect [39]. In this study, we investigated the effects of NTAPP using a clinically relevant rat model of normal and non-union fracture models, and an in vitro study of MC3T3-E1 cells, which are known to continuously express osteoblastic factors similar to those of bone formation in vivo [54].

For the in vivo study, we irradiated a normal fracture model with NTAPP. In this model, every group showed bone healing after 4 weeks, and we could not detect a treatment-specific effect of NTAPP. However, this also demonstrated that NTAPP irradiation for 5–15 min was not harmful and did not hinder the process of bone-union. Next, we investigated whether irradiation for a short-term affected the extracellular matrix, using XPS and WCA. It was revealed that a 5 min irradiation changes both the surface chemistry and surface wettability of the bone surface. Based on these results, a 5 min irradiation that affects the bone surface within a safe range was adopted for studying the non-union fracture model. The results showed that NTAPP irradiation of the non-union fracture model promoted bone healing and improved the biomechanical properties. In addition, the histological evaluation showed that Masson's trichrome staining had very few calluses in the control group. In contrast, rich calluses were observed in the NTAPP-treated group, indicating continued progression of normal bone healing. Safranin-O staining revealed endochondral ossification around the fracture site.

Fracture healing occurs when mesenchymal stem cells (MSCs) differentiate into osteoblasts and chondrocytes. Further, stem cells are recruited from the periosteum, muscle tissue, and bone marrow. Because the periosteum is cauterized in the non-union model, stem cells necessary for fracture healing are recruited primarily from the environment surrounding the

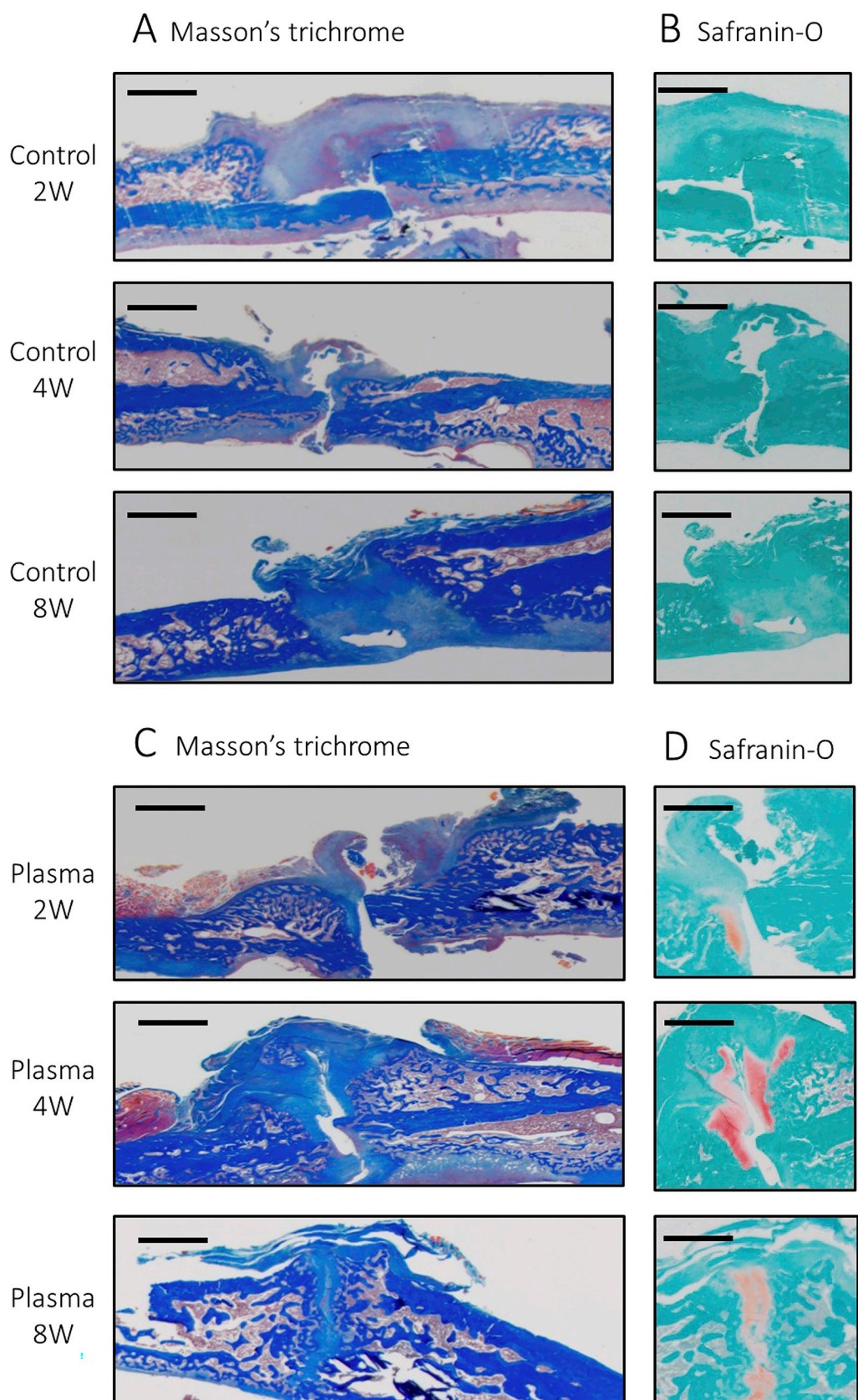

**Fig 5.** Tissue sections of non-union fracture models without plasma treatment, **A** shows Masson's trichrome staining, and **B** shows Safranin-O staining at 2, 4, and 8 weeks after the surgery. Tissue sections of non-union fracture models with plasma treatment, **C** shows Masson's trichrome staining, and **D** shows Safranin-O staining at 2, 4, and 8 weeks after the surgery. The scale bar indicates 500 μm.

fracture site, including the muscle tissue and bone marrow. The control group did not show adequate bone healing. However, the NTAPP-treated group initiated endochondral ossification, which eventually progressed to bone healing. This could be because the control group did not achieve bone healing as they did not have enough stem cells recruited. In contrast, the NTAPP-treated group effectively recruited and proliferated stem cells from the muscle tissue and bone marrow near the fracture site affected by NTAPP. This process may have contributed to the successful maintenance of bone union. Moriguchi et al. reported that NTAPP improves the wettability of artificial bone consisting of hydroxyapatite (HA) and increases osteoconductivity [55]. As we applied NTAPP only during the initial operation for 5 min, this effect of NTAPP on bone material to improve osteoconductivity may be advantageous in the early phase of the recruitment of undifferentiated mesenchymal cells.

Furthermore, fracture healing proceeds in five sequential temporal phases: (i) hematoma formation, (ii) inflammation, (iii) angiogenesis, (iv) cartilage formation, and (v) bone remodeling [56, 57]. Several studies have shown that NTAPP affects angiogenesis [58–60] and modulates inflammation [60–63]. Duchesne et al. found that NTAPP modulates endothelial nitric oxide synthase (eNOS) signaling and promotes angiogenesis [59]. In another study by Tan et al., NTAPP promoted the angiogenesis of the rat cutaneous wound model by upregulating proangiogenic markers CD31, VEGF, and TGF-β [60]. Regarding modulating inflammation, Liu et al. reported that NTAPP activated nuclear transcription factor κB (NF-κB), which regulate, immune response, and inflammation [62]. Paola et al., further found that the PPAR-c anti-inflammatory molecular pathway was activated by NTAPP [63]. Because angiogenesis and inflammatory control are important factors in bone healing, they should be further investigated in the future, using in vivo fracture models. NTAPP may affect some of the bone-healing processes.

We conducted an in vitro study to test the effect of NTAPP on the cell viability and differentiation of MC3T3-E1 cells treated with the same device used in animal experiments to investigate the influence of NTAPP on osteoblasts. Regarding cell viability, no significant difference was observed in the 24 h and 48 groups after NTAPP irradiating for 5–15 s.

These non-toxic doses were then used for the cell differentiation assays. To evaluate the effect of NTAPP on cell differentiation, we measured the activity of ALP, a marker of osteoblast differentiation, in MC3T3-E1 cells. In this protocol, since medium exchange is necessary, NTAPP was irradiated every two days simultaneously with medium exchange to maintain the culture environment in the medium treated with NTAPP. The results showed that irradiation for 5–15 s significantly increased ALP activity compared to the positive control. There was no significant difference in the ALP activity between the groups treated with differing durations of NTAPP irradiation, suggesting that even a short irradiation duration of 5 s may stimulate cell differentiation. Regarding short-term irradiation, Tominami et al. demonstrated that even 5 s of irradiation could facilitate reactions between radicals and the culture medium [64]. In their work, they argued that chemical species generated secondarily by the reaction between the radicals and the culture medium stimulated osteoblastic differentiation of the cells.

The diverse effects of NTAPP irradiation can be attributed to its complex chemical composition, which is influenced by various conditions under which it is produced. It has long been reported that reactive oxygen and nitrogen species (RONS) generated by NTAPP play a key role in cellular responses in vitro and in vivo [29]. Reactive oxygen and nitrogen species

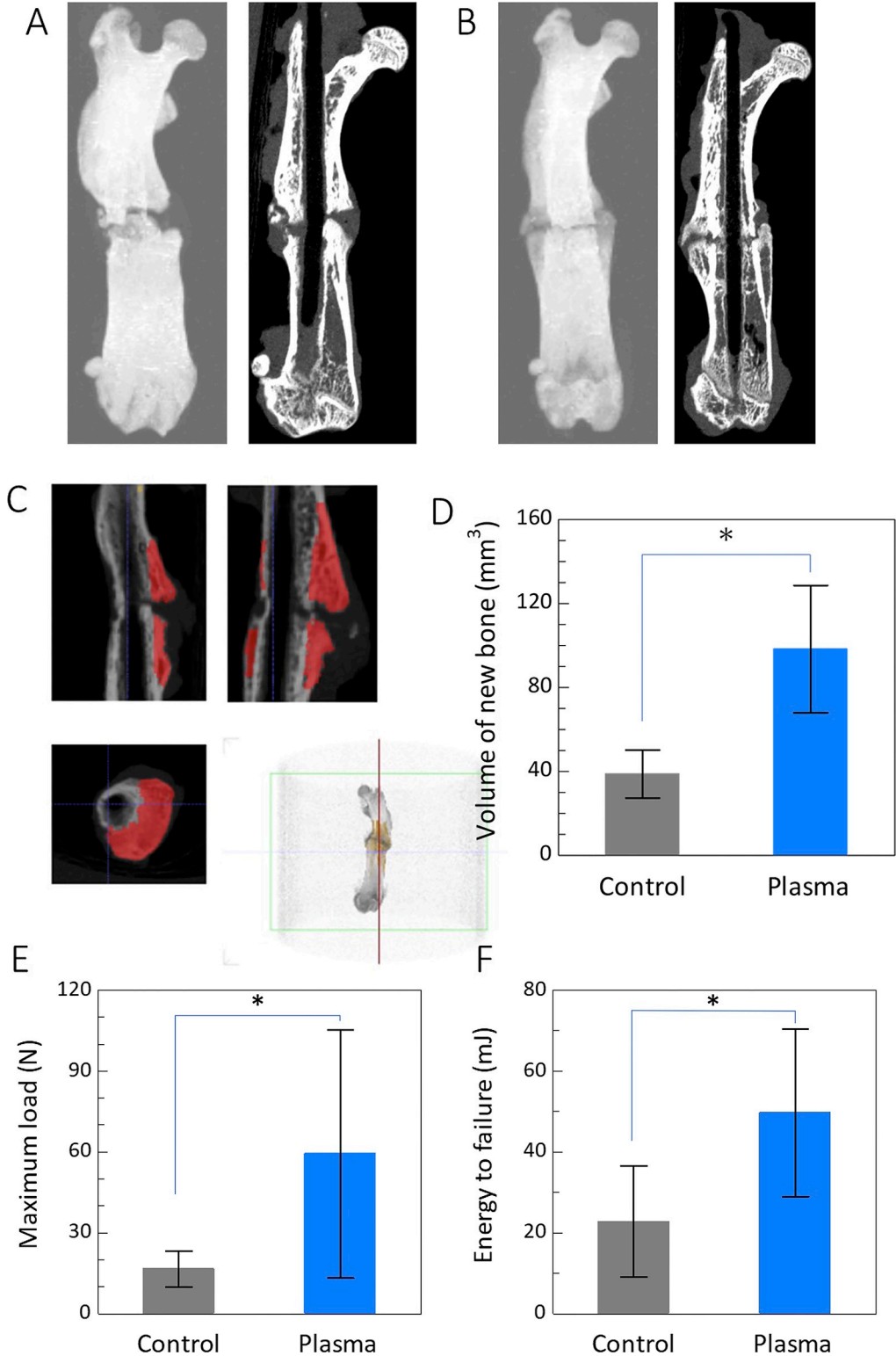

**Fig 6. A** and **B** show representative images (left image of each pair) of 3D-reconstructed μ-CT and coronal plane views (right image of each pair) for untreated control and plasma treated non-union fracture models, respectively, at 8 weeks after the surgery. **C** shows new bone volume calculated using a 3D image processing software (ExFact VR; Nihon Virtual Science). **D** shows the bar graph illustrating the new bone volume comparing the control with NTAPP-treated group. **E** shows the maximum load, and **F** shows the failure energy of the femur of rats with and without NTAPP treatment at 8 weeks after the surgery. Mechanical properties were improved after the NTAPP treatment.

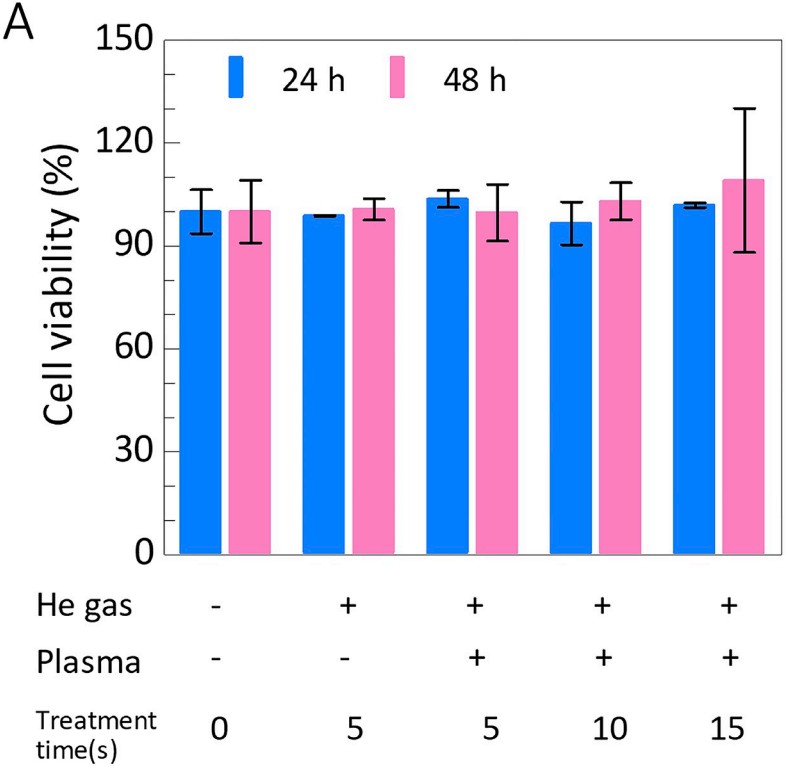

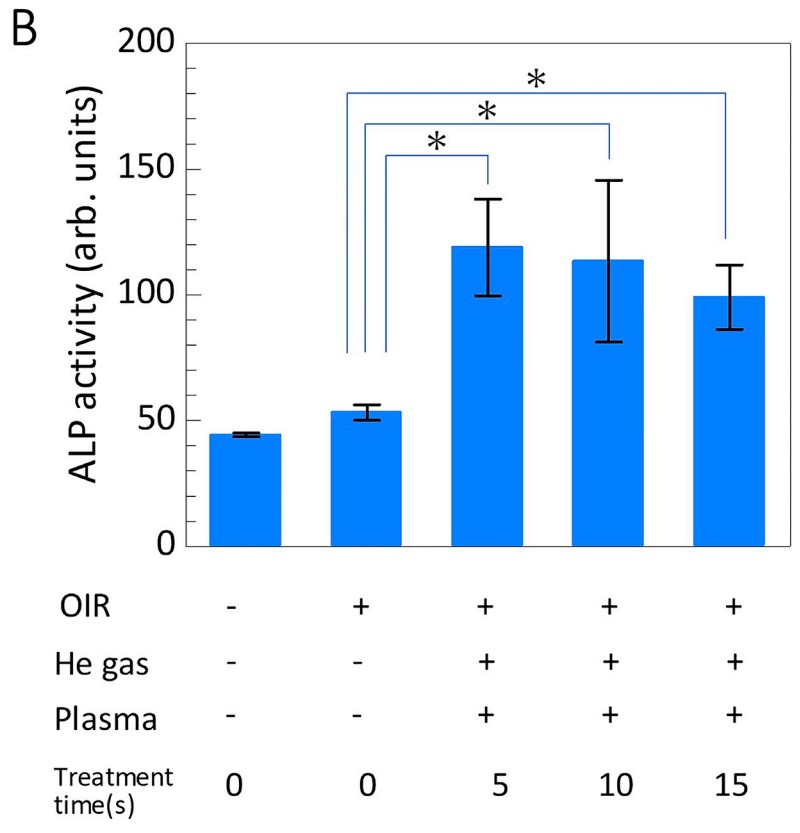

**Fig 7. A** shows the MC3T3-E1 cell viability at 24 h and 48 h in cases of with and without NTAPP treatment up to 15 s. There is no significant cell viability after the NTAPP treatment. Alkaline phosphatase (ALP) activity was increased up to almost double in all cases with NTAPP treatment.

(RONS) include superoxide anion ($O_2^-$), hydroxyl radical ($OH^{\bullet}$), hydrogen peroxide ($H_2O_2$), nitric oxide (NO), peroxynitrite ($ONOO^-$), and other related compounds originating from cellular metabolic processes and the external environment [62]. Normal levels of RONS play crucial roles in signal transduction in organisms. Although a higher dose of RONS induces apoptosis and necrosis [65]. However, we did not observe any impairment in the proliferation of MC3T3-E1 cells. Liu et al. reported that the production of intracellular RONS increased with prolonged NTAPP treatment time [62]. Therefore, the irradiation time used in this study may have been appropriate, without affecting the cell viability. Furthermore, increased ALP activity after multiple irradiations promotes early osteoblast differentiation.

This study had several limitations. First, we did not evaluate the potential long-term effects of NTAPP in this study. However, to the best of our knowledge, no serious adverse effects of prolonged irradiation have been reported in the literature. Several studies have evaluated the effects of NTAPP on major organs in animal models and have reported no significant safety concerns [66–68]. However, Najafzadehvarz et al. reported prolonged irradiation time and shortened NTAPP distance in certain modes, resulted in epithelial tissue damage in animal models [69]. The optimal conditions for NTAPP use and its long-term effects must, therefore, be carefully investigated. Second, the basic mechanism by which NTAPP promotes bone healing in a non-union fracture model remains unknown. This study showed that callus formation was triggered by endochondral ossification in vivo and promoted cell differentiation in vitro. Further gene-level analyses may shed more light on this mechanism. Third, this study used MC3T3-E1 cells, which are immortalized cells; MC3T3-E1 cells have been commonly used in many studies for studying osteoblasts, but primary cells would be more relevant and a better model for future use.

Finally, NTAPP is composed of multiple components, including electrically neutral RONS, charged species, and high-energy photons [29, 70], and the key component that promotes bone healing is unclear. However, these factors are expected to be comprehensively involved in the environment where NTAPP is applied. Nevertheless, this study confirmed the bone-regenerative potential of NTAPP irradiation both in vivo and in vitro. These results may contribute to the development of fracture treatments, including fracture non-unions. In the future, genetic analyses to elucidate the inherent mechanisms promoting bone healing and evaluation of the effect of NTAPP on mesenchymal stem cells are needed.

## Supporting information

**S1 Data.**
(XLSX)

## Acknowledgments

We would like to thank Editage (www.editage.jp) for English language editing.

## Author Contributions

**Conceptualization:** Kosuke Saito, Hiromitsu Toyoda, Tatsuru Shirafuji, Hiroaki Nakamura.

**Data curation:** Kosuke Saito, Hiromitsu Toyoda, Mitsuhiro Okada, Katsumasa Nakazawa, Yoshitaka Ban, Kumi Orita, Akiyoshi Shimatani, Hana Yao.

**Formal analysis:** Kosuke Saito, Mitsuhiro Okada, Jun-Seok Oh, Katsumasa Nakazawa, Yoshi-taka Ban, Kumi Orita, Akiyoshi Shimatani, Hana Yao.

**Funding acquisition:** Hiromitsu Toyoda, Jun-Seok Oh, Tatsuru Shirafuji, Hiroaki Nakamura.

**Investigation:** Kosuke Saito, Hiromitsu Toyoda, Jun-Seok Oh, Tatsuru Shirafuji.

**Methodology:** Kosuke Saito, Hiromitsu Toyoda, Hiroaki Nakamura.

**Project administration:** Hiromitsu Toyoda, Tatsuru Shirafuji, Hiroaki Nakamura.

**Supervision:** Hiromitsu Toyoda, Jun-Seok Oh, Tatsuru Shirafuji, Hiroaki Nakamura.

**Validation:** Kosuke Saito, Hiromitsu Toyoda, Mitsuhiro Okada, Jun-Seok Oh, Katsumasa Nakazawa, Yoshitaka Ban, Kumi Orita, Akiyoshi Shimatani, Hana Yao.

**Visualization:** Kosuke Saito, Mitsuhiro Okada, Jun-Seok Oh, Katsumasa Nakazawa, Yoshitaka Ban, Akiyoshi Shimatani, Hana Yao.

**Writing – original draft:** Kosuke Saito, Hiromitsu Toyoda, Jun-Seok Oh, Kumi Orita.

**Writing – review & editing:** Kosuke Saito, Hiromitsu Toyoda, Jun-Seok Oh, Tatsuru Shirafuji, Hiroaki Nakamura.

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
