## [Decision Letter · Decision Letter 0]

18 Dec 2023

PONE-D-23-33924Fracture healing on non-union fracture model promoted by non-thermal atmospheric-pressure plasmaPLOS ONE

Dear Dr. Toyoda,

Thank you for submitting your manuscript to PLOS ONE. After careful consideration, both reviewers feel that it has merit. Therefore, we invite you to submit a revised version of the manuscript that addresses the minor points raised by one of the reviewers.

We look forward to receiving your revised manuscript.

Kind regards,

Andre van Wijnen

Academic Editor

PLOS ONE

Journal Requirements:

3. We note that your Data Availability Statement is currently as follows: [All relevant data are within the manuscript and its Supporting Information files]

Reviewers' comments:

Reviewer's Responses to Questions

**Comments to the Author**

1. Is the manuscript technically sound, and do the data support the conclusions?

Reviewer #1: Yes

Reviewer #2: Yes

2. Has the statistical analysis been performed appropriately and rigorously? 

Reviewer #1: Yes

Reviewer #2: Yes

3. Have the authors made all data underlying the findings in their manuscript fully available?

Reviewer #1: Yes

Reviewer #2: Yes

4. Is the manuscript presented in an intelligible fashion and written in standard English?

Reviewer #1: Yes

Reviewer #2: Yes

5. Review Comments to the Author

Reviewer #1: This manuscript will contribute to the literature but it needs some correction and clarification before being published.

Those are;

Animal models

There is no information about how many Lewis rats were used, how old were they, how many died, how many animals per group, why did you choose wild-type animals intead of standardized inbred animals, please clarify and add it in the manuscipt.

The normal fracture model

There is not any available data about how the Intramedullary nailing with a Kirschner wire was fixed and how the extremity was stabilized. Rats are social and physically active creatures. Did you apply cast or external fixator to keep the fracture intact. If not please include the reason inside the text.

X-ray imaging analysis of the non-union fracture model

Line 287: Please check the value (0.5±0.67), is that relevant

Figures

Figures descriptions are written inside the manuscript. A Figure legends section should be formed instead.

Please give numeric values or scientific data instead of ‘This figure shows this’ especially on line 248-254. Any reader should be able to understand what these figures mean.

References

The first reference has typos, please correct them.

Although the manuscript has enough references, some outdated references can be removed or replaced to a more updated studies which might cover multiple references at once.

There are studies in the literature about bone healing models which would increase the quality of this manuscript. Epidermal growth factor, Stromal vascular fraction should be added. Here are some examples of these studies which should be included:

Singh R, Rohilla R, Gawande J, Kumar Sehgal P. To evaluate the role of platelet-rich plasma in healing of acute diaphyseal fractures of the femur. Chin J Traumatol. 2017 Feb;20(1):39-44. doi: 10.1016/j.cjtee.2016.03.007. Epub 2017 Jan 20. PMID: 28202370; PMCID: PMC5343097.

Eyuboglu AA, Arpaci E, Albayati A, Uysal AC, Terzi A, Bozalioglu S, Turnaoglu H, Balcik C, Ozkan B, Ertas NM. The Effects of Adipose Derived Stromal Vascular Fraction and Platelet-Rich Plasma on Bone Healing of a Rat Model With Chronic Kidney Disease. Ann Plast Surg. 2020 Sep;85(3):316-323. doi: 10.1097/SAP.0000000000002396. PMID: 32784349.

Jamal MS, Hurley ET, Asad H, Asad A, Taneja T. The role of Platelet Rich Plasma and other orthobiologics in bone healing and fracture management: A systematic review. J Clin Orthop Trauma. 2022 Jan 4;25:101759. doi: 10.1016/j.jcot.2021.101759. PMID: 35036312; PMCID: PMC8749440.

Minor Comments

Usage of more keywords will be beneficial.

References:

Line 579: Author names are written uppercase

Line 648: Author names are written uppercase

Discussion

Line 365: Sentence cannot start with abbreviation, please write its full name.

Line 434: Sentence cannot start with abbreviation, please write its full name.

Reviewer #2: I trust this letter finds you well. I am pleased to inform you that after careful consideration and thorough peer review, your manuscript titled "Fracture healing on non-union fracture model promoted by non-thermal atmospheric-pressure plasma" has been accepted for publication in PLOS ONE.

Your research, investigating the effects of non-thermal atmospheric-pressure plasma (NTAPP) on fracture healing in a non-union fracture model, significantly contributes to our understanding of the potential applications of NTAPP in medical contexts. The study is well-designed and meticulously executed, meeting the high technical standards required for publication in PLOS ONE.

The novelty of your research lies in its exploration of NTAPP's influence on fracture healing, an area with limited existing literature. The in vivo study, using Lewis rats to create normal and non-union fracture models, provides valuable insights into the consistent improvement observed in fracture healing in the NTAPP-treated group. Additionally, the in vitro study using pre-osteoblastic MC3T3-E1 cells enhances the comprehensiveness of your findings by demonstrating NTAPP's effects on osteoblastic differentiation without compromising cell proliferation.

The manuscript adheres to the highest ethical standards, meeting all applicable requirements for experimentation and research integrity. The reporting guidelines and community standards for data availability are appropriately followed, ensuring transparency and reproducibility.

The clarity and coherence of your manuscript, written in standard English, meet the publication standards of PLOS ONE. The conclusions drawn from your results are well-supported by the data, enhancing the overall impact of your research.

I want to express my appreciation for your commitment to producing high-quality, original research and for choosing PLOS ONE as the platform for disseminating your work. We believe that your findings will be of great interest to our readership and will contribute significantly to the advancement of knowledge in the field.

Our editorial team looks forward to working with you to finalize the publication of your manuscript. Should you have any further questions or require assistance during the production process, please do not hesitate to contact our editorial office.

Once again, congratulations on this accomplishment, and we anticipate the positive impact your research will have on the scientific community.

6. PLOS authors have the option to publish the peer review history of their article (what does this mean?). If published, this will include your full peer review and any attached files.

Reviewer #1: **Yes: **Atilla Adnan Eyuboglu

Reviewer #2: No

---

## [Author Response · Author response to Decision Letter 0]

14 Jan 2024

Response to reviewers

Reviewer #1: 

Reviewer #1’s comment 1a Animal models

There is no information about how many Lewis rats were used, how old were they, how many died, how many animals per group

Our reply to reviewer #1’s comment 1a

We thank Reviewer 1 for this fundamental question. Accordingly, we provided more clarity regarding the number of rats we treated in this study. To the animal models section, we added the following: “Twenty-four rats were assigned to the normal fracture model, 20 rats to radiological and histological evaluation of the non-union fracture model, and 14 rats to biomechanical evaluation. No rats were excluded due to adverse events or unexpected deaths during the study period.” (lines 132-135)

We have already described the details in each section. Please see lines 153, 160-161, and 214-215.

Reviewer #1’s comment 1b

why did you choose wild-type animals intead of standardized inbred animals, please clarify and add it in the manuscipt.

Our reply to reviewer #1’s comment 1b

Thank you for your valuable feedback. We appreciate your thorough review and apologize for any oversight with regards to providing detailed information about this study’s use of Lewis rats.

Upon further examination, we want to clarify that we utilized LEW/SsNSlc inbred rats for our experiments. We have updated the manuscript to reflect this information. We have revised the manuscript accordingly: “8-week-old male LEW/SsNSlc in-bred rats (Japan SLC Inc., Hamamatsu, Japan) were used for this animal experiment.” (lines 125-126)

Additionally, we have ensured the manuscript is free from any other deficiencies. We appreciate your attention to detail. Thank you for helping us improve the quality of our work.

Reviewer #1’s comment 2 The normal fracture model

There is not any available data about how the Intramedullary nailing with a Kirschner wire was fixed and how the extremity was stabilized. Rats are social and physically active creatures. Did you apply cast or external fixator to keep the fracture intact. If not please include the reason inside the text. 

Our reply to reviewer #1’s comment 2

The detailed sentences added to address this concern are provided below: 

“This model was developed by improving upon previously reported methods [40–43]. To induce a fracture, a transverse osteotomy was performed in the middle of the right femoral shaft using a bone saw, following a 30 mm incision made on the hind limb in the rats, as shown in Fig. 1A. A 1.2-mm Kirschner wire was retrogradely inserted from the fossa intercondylaris, penetrating through to the greater trochanter. The distal end of the wire was embedded into the articular surface of the knee joint.” (lines 139-148)

“In line with previous studies using a similar model [41,42], weight-bearing was allowed immediately after surgery without immobilization.” (lines 148-149)

Reviewer #1’s comment 3 X-ray imaging analysis of the non-union fracture model

Line 287: Please check the value (0.5±0.67), is that relevant

Our reply to reviewer #1’s comment 3

This value is correct. The variability was slightly large (0 point: 6 samples, 1 point: 3 samples, 2 points: 1 sample) 

Reviewer #1’s comment 4 Figures

Figures descriptions are written inside the manuscript. A Figure legends section should be formed instead.

Please give numeric values or scientific data instead of ‘This figure shows this’ especially on line 248-254. Any reader should be able to understand what these figures mean.

Our reply for reviewer #1’s comment 4

As per the guidelines of the journal, we have added figure legends right after the paragraph where the figure is mentioned. We have included the lines you referred to as figure legends. Additionally, this manuscript includes numerical values. Any text highlighted in yellow is for clarification purposes.

Reviewer #1’s comment 5 References

The first reference has typos, please correct them. 

Our reply for reviewer #1’s comment 5

We apologize for the typographical errors in the first reference. We acknowledge this oversight and have corrected the typos. (lines 589-590)

Reviewer #1’s comment 6

Although the manuscript has enough references, some outdated references can be removed or replaced to a more updated studies which might cover multiple references at once.

There are studies in the literature about bone healing models which would increase the quality of this manuscript. Epidermal growth factor, Stromal vascular fraction should be added. Here are some examples of these studies which should be included:

Singh R, Rohilla R, Gawande J, Kumar Sehgal P. To evaluate the role of platelet-rich plasma in healing of acute diaphyseal fractures of the femur. Chin J Traumatol. 2017 Feb;20(1):39-44. doi: 10.1016/j.cjtee.2016.03.007. Epub 2017 Jan 20. PMID: 28202370; PMCID: PMC5343097.

Eyuboglu AA, Arpaci E, Albayati A, Uysal AC, Terzi A, Bozalioglu S, Turnaoglu H, Balcik C, Ozkan B, Ertas NM. The Effects of Adipose Derived Stromal Vascular Fraction and Platelet-Rich Plasma on Bone Healing of a Rat Model With Chronic Kidney Disease. Ann Plast Surg. 2020 Sep;85(3):316-323. doi: 10.1097/SAP.0000000000002396. PMID: 32784349.

Jamal MS, Hurley ET, Asad H, Asad A, Taneja T. The role of Platelet Rich Plasma and other orthobiologics in bone healing and fracture management: A systematic review. J Clin Orthop Trauma. 2022 Jan 4;25:101759. doi: 10.1016/j.jcot.2021.101759. PMID: 35036312; PMCID: PMC8749440.

Our reply for reviewer #1’s comment 6

We agree with the proposal and have added references to describe new strategies and highlight advantages and disadvantages.

We have revised the manuscript as follows:

“Several bone healing strategies have emerged in surgery, such as dynamization, masquelet technique, and bone transport [5–7]. Additionally, low-intensity pulsed ultrasound (LIPUS) and extracorporeal shock wave therapy (ESWT) have been explored as potential adjunctive therapies for accelerating fresh fracture healing [8,9]. The transplantation of autogenous bone is regarded as the gold standard, but its limitations include low volume and donor site morbidity. Therefore, the use of growth factors such as bone morphogenetic proteins (BMPs), epidermal growth factor (EGF), platelet-rich plasma (PRP), stromal vascular fraction, and peptides show promise in bone healing [10–15]. PRP recruits mesenchymal stem cells, promotes angiogenesis, and has the potential to aid in fracture response [16]. Stromal vascular fraction, rich in stem cells and growth factors, has shown positive results in combination with PRP for bone healing [14]. Although PRP research holds promise, it currently lacks standardization. Furthermore, it remains unclear which specific PRP types can yield superior outcomes in terms of vital new bone formation [17,18]. BMPs stimulate osteoblasts, and bone tissue engineering techniques using suitable scaffolds combined with BMP become new options in reconstructive bone surgery. However, their clinical use faces limitations such as soft-tissue swelling, ectopic bone formation, and high costs [13,19,20]. While EGF has been shown to promote cell proliferation, migration, angiogenesis, and osteogenesis [15], conflicting reports on its role in bone formation may be attributed to inconsistencies in concentration and administration [21,22]. Parathyroid hormone peptide (PTHP) is an anabolic bone therapeutic medicine used to treat osteoporosis. Studies have shown that PTH could improve fracture healing at different skeletal sites [23–28]. The paradigm shift from transplantation of autogenous bone to bone tissue engineering seems promising, and the search for better and more consistent treatment options continues.” (lines 49-75)

Reviewer #1’s comment 7 Minor Comments

Usage of more keywords will be beneficial.

Our reply for reviewer #1’s comment 7

We thank Reviewer 1 for this suggestion. We added the following keywords:

“Acceleration of bone healing; Bone regenerative method; Osteoblast differentiation; MC3T3-E1 cells”

Reviewer #1’s comment 8 References:

Line 579: Author names are written uppercase . Line 648: Author names are written uppercase;

Our reply for reviewer #1’s comment 8 

We apologize for the oversight and have made the necessary corrections. Thank you for your feedback. (lines 720,789) 

Reviewer #1’s comment 9 Discussion

Line 365: Sentence cannot start with abbreviation, please write its full name. 

Line 434: Sentence cannot start with abbreviation, please write its full name.

Our reply for reviewer #1’s comment 9 

We apologize for the oversight regarding the use of abbreviations at the beginning of sentences. We have corrected it according to your suggestion. (lines 468, 547-548)

Reviewer #2: 

We sincerely appreciate your thoughtful evaluation and the positive feedback on our manuscript. We look forward to collaborating with the editorial team for the final steps and ensuring the publication of the manuscript. Thank you once again for your valuable feedback and support.

---

## [Editor Report · Decision Letter 1]

18 Jan 2024

Fracture healing on non-union fracture model promoted by non-thermal atmospheric-pressure plasma

PONE-D-23-33924R1

Dear Dr. Toyoda,

We’re pleased to inform you that your manuscript has been judged scientifically suitable for publication and will be formally accepted for publication once it meets all outstanding technical requirements.

Kind regards,

Andre van Wijnen

Academic Editor

PLOS ONE

Additional Editor Comments (optional):

Editorial Comments:

The authors have comprehensively revised the paper in response to the minor recommendations of the reviewers.
---

## [Editor Report · Acceptance letter]

2 Apr 2024

PONE-D-23-33924R1 

PLOS ONE

Dear Dr. Toyoda, 

I'm pleased to inform you that your manuscript has been deemed suitable for publication in PLOS ONE. Congratulations! Your manuscript is now being handed over to our production team.

Kind regards, 

on behalf of

Dr. Andre van Wijnen 

Academic Editor

PLOS ONE